# Factors Influencing the Degree of Gastric Atrophy in *Helicobacter pylori* Eradication Patients with Drinking Habits

**DOI:** 10.3390/microorganisms12071398

**Published:** 2024-07-10

**Authors:** Kayoko Ozeki, Kazuhiro Hada, Yoshifumi Wakiya

**Affiliations:** 1Laboratory of Pharmacy Practice and Sciences, Aichi Gakuin University, Nagoya 4648650, Aichi, Japan; khada@dpc.agu.ac.jp (K.H.); y-wakiya@dpc.agu.ac.jp (Y.W.); 2Department of Community Health and Preventive Medicine, Hamamatsu University School of Medicine, Hamamatsu 4313192, Shizuoka, Japan

**Keywords:** *Helicobacter pylori*, eradication patients, drinking habits

## Abstract

Chronic gastritis caused by *Helicobacter pylori* (*H. pylori*) infection can lead to gastric atrophy. This study aimed to identify the factors associated with gastric atrophy in *H. pylori* eradication patients with drinking habits. Of the 250 patients who visited Hamamatsu University Hospital for *H. pylori* eradication and underwent eradication treatment between April 2017 and December 2020, 127 patients with drinking habits were included in this study. The degree of gastric atrophy of the patients was classified based on endoscopy. The relationship between patient attributes (sex, age, alcohol consumption, frequency of drinking, smoking status, and medication use) and a highly atrophic stomach was statistically analyzed. The results showed that gastric atrophy was significantly higher in males and in those aged 60 years or older and that gastric atrophy tended to be higher in those who drank 20 g or more per day and 5 days or more a week. There was also a trend toward higher atrophy in sake drinkers and lower atrophy in wine drinkers. This study provides useful knowledge for patient management and guidance after *H. pylori* eradication treatment and indicates the importance of comprehensive measures, including alcohol consumption control and lifestyle modification, especially for men and older people.

## 1. Introduction

*Helicobacter pylori* (*H. pylori*) is a bacterium that lives in the gastric mucosa, and prolonged infection increases the risk of developing chronic gastritis, gastric ulcers, and gastric cancer [1,2,3,4,5]. When *H. pylori* infect the gastric mucosa, inflammation occurs, and over time, the atrophic mucosa gradually spreads, leading to atrophic gastritis, a condition in which the gastric mucosa becomes unevenly thin and atrophic [6]. This condition is widely recognized as a precursor lesion to gastric cancer and increases the risk of gastric cancer [7,8], so early diagnosis and treatment are important. The eradication of *H. pylori* (treatment with antibiotics) has been shown to suppress the inflammation of the gastric mucosa and delay the progression of atrophy, thereby reducing the risk of gastric cancer development [9].

However, even with various treatment strategies for eradication therapy, such as antibiotic combinations and changes in the number of treatment days, some patients are known to have difficulty in eradication [10,11]. In a previous study, the authors surveyed patients who came for *H. pylori* eradication therapy and found that allergies and drinking habits were associated with difficulty in eradication [12,13,14,15]. A meta-analysis also found that in an Asian population, higher daily alcohol consumption was associated with a higher risk of *H. pylori* eradication failure [16].

In this study, we focused on the drinking habits of Japanese patients who visited a hospital for *H. pylori* eradication, selected only those patients with a drinking habit among those known to have *H. pylori* infection, and aimed to examine the factors related to the degree of gastric atrophy in these patients.

## 2. Materials and Methods

### 2.1. Subjects and Study Procedure

Data on gastric atrophy levels were obtained from completed questionnaires and endoscopic examinations of 250 patients who visited the outpatient clinic specializing in *H. pylori* eradication at Hamamatsu University Hospital for *H. pylori* eradication from April 2017 to December 2020. Six levels of gastric atrophy were diagnosed, ranging from C-1 to O-3. The levels were defined as follows: C-1 (atrophy near the gastric outlet (pylorus)), C-2 (atrophy from C-1 to the lower 1/3 of the stomach), C-3 (atrophy from C-2 to the lower 2/3 of the stomach), O-1 (atrophy near the gastric inlet (fenestra)), O-2 (atrophy to the stomach wall (fenestra)), and O-3 (total atrophy and loss of folds). The eradication treatment for the patients involved first-line therapy with amoxicillin, clarithromycin, and vonoprazan. The second-line therapy included amoxicillin, metronidazole, and vonoprazan. After the third-line therapy, the choice of drug was left to the discretion of the physician, with sitafloxacin and minocycline often replacing amoxicillin. Patients were informed of the purpose of the study, and their consent was obtained. The questionnaire items included age, gender, alcohol consumption, alcohol intake, frequency of alcohol consumption, smoking status, and current medications. Of these patients, those with autoimmune gastritis were excluded, 127 with a drinking habit were included in the study, and those who could not respond to the required items were excluded.

This study was approved by the Medical Ethics Committee of Hamamatsu University School of Medicine (No. 17-072).

### 2.2. Statistical Analysis

Simple tabulations were performed for the following attributes: sex, age, amount of alcohol consumed per day, frequency of drinking, smoking status, medication use, and degree of gastric atrophy.

Cross-tabulations were performed to examine the relationship between each of the patient attributes and the level of gastric atrophy. The percentage of patients with high or low gastric atrophy by type of alcohol (sake, beer, shochu, wine, other) was also tabulated. The alcohol content is approximately 11~15% for sake, 4~8% for beer, 25% for shochu, and 8~12% for wine. The alcoholic beverages equivalent to 20 g of alcohol are as follows: sake (15%: 180 mL), beer (5%: 500 mL), shochu (25%: 100 mL), and wine (14%: 200 mL). For beer, which had the highest number of drinkers by type, atrophy levels were further divided into two categories: those drinking 20 g of alcohol or more and those drinking less than 20 g of alcohol.

Logistic regression analysis was conducted using each of the patient attributes as an explanatory variable and a highly atrophic stomach as the objective variable. Highly atrophic stomachs were defined as O1, O2, and O3, and stomachs with a low level of atrophy were defined as C1, C2, and C3. Model 1 included sex and age (<60 years and ≥60 years) as explanatory variables, Model 2 added alcohol consumption per day (<20 g and ≥20 g) as a covariate, and Model 3 added alcohol consumption per day (<20 g and ≥20 g), smoking status, and taking medication as covariates. Statistical analysis was performed with JMP13.

## 3. Results

The demographic characteristics of *H. pylori* eradication patients with drinking habits are shown in Table 1. Males and females accounted for 51.2% and 48.8%, respectively, and many patients were in their 50s and 60s (in the 20% range). Of the total patients, 65.4% consumed less than 20 g of alcohol per day, and 18.1% consumed 20 g of alcohol or more. This included 73.9% (17/23) of men and 26.1% (6/23) of women consuming 20 g or more of alcohol. The frequency of alcohol consumption was 34.6% for 5 or more days per week. There were few smokers (8.7%). Male smokers accounted for 72.7% (8/11) and female smokers for 27.3% (3/11). Of the total patients, 63.0% were taking medications. The medications taken included antihypertensive drugs, anticoagulants, gastroprotective agents, non-steroidal anti-inflammatory drugs, lipid-lowering agents, anti-anxiety and sleep medications, antidiabetic agents, and other drugs. The patients’ gastric atrophy was high in the 20% range for C2 and C3, which are relatively low atrophic states.

Table 2 shows the cross-tabulation results of each of the patient attributes with drinking habits and high/low gastric atrophy. Males and those aged 60 years and older had significantly higher levels of gastric atrophy. The degree of gastric atrophy tended to be higher in patients who consumed 20 g or more of alcohol per day, drank 5 days or more per week, and took medications.

Table 3 shows high and low gastric atrophy by the type of alcohol consumed. The average overall high atrophy was 35.9%. Sake drinkers had a 17.1% higher atrophy than average. Conversely, the percentage of high atrophy in wine drinkers was 14.5% lower than average.

Table 4 shows the results of a logistic regression analysis of the association between each of the attributes of *H. pylori* eradication patients with drinking habits as explanatory variables and a highly atrophic stomach as the objective variable. In Model 1, in which the explanatory variables gender and age were entered simultaneously, the odds ratios of 3.18 (95% confidence interval 1.22–8.87) and 2.88 (1.34–7.51) were significantly higher for males and patients aged 60 years and older, respectively. In Model 2, which adjusted for alcohol consumption as a covariate, atrophy was significantly higher in patients aged 60 years and older, with an odds ratio of 2.81 (1.01–8.07), and in Model 3, which adjusted for alcohol consumption, smoking status, and drug use as covariates, atrophy was significantly higher in men, with an odds ratio of 3.43 (1.08–11.99).

## 4. Discussion

In this study, we identified the factors influencing the degree of gastric atrophy in *H. pylori*-infected patients with a drinking habit. While previous studies have examined the impact of either *H. pylori* infection or alcohol consumption on gastric atrophy separately [17,18], few studies have investigated the combined effect of these factors. By focusing on patients undergoing *H. pylori* eradication therapy, who also consume alcohol, our study aimed to provide additional insights into the factors associated with gastric atrophy, offering a more comprehensive understanding of its etiology in this specific patient population.

The results of this study revealed that among *H. pylori*-infected patients with drinking habits, the degree of gastric atrophy was higher in men and in those aged 60 years or older. The reasons for the higher degree of atrophy in men are the presence of the miR-27a polymorphism, which has been reported to be associated with gastric mucosal atrophy only in men [19], and patients with high serum testosterone levels as an effect of male hormones are reported to be at higher risk for chronic atrophic gastritis [20]. Regarding age, it has been consistently observed that gastric atrophy is more pronounced in older people [21,22]. The prevalence of gastric atrophy escalates with seniority, as evidenced by the literature reporting a 50% decline in patients with normal gastric mucosa over their lifespan, especially in regions with high *H. pylori* prevalence [23]. Furthermore, an animal study has corroborated the strong correlation between gastric mucosal atrophy and cellular senescence [24]. These findings are consistent with previous studies and reaffirm that age and gender are major factors influencing gastric atrophy, even among those with drinking habits and *H. pylori* infection.

This study distinguishes itself by examining the unique aspects of *H. pylori* infection and drinking habits and their direct impact on gastric atrophy. Regarding alcohol consumption, the results of the cross-tabulation showed that gastric atrophy tended to be higher in those who consumed 20 g or more of alcohol per day and drank 5 days per week or more. Logistic regression analysis showed that for patients aged 60 years and older, in Model 2, an alcohol intake of 20 g or more per day was significantly associated with a higher degree of gastric atrophy.

Previous studies have reported conflicting findings regarding the association between alcohol consumption and gastric atrophy [22]. Regarding gastric cancer, for which atrophic gastritis is sometimes considered a precursor symptom, one study reported that alcohol consumption is an important risk factor associated with an increased incidence of gastric cancer in a non-*H. pylori*-infected population and found no significant association between alcohol consumption and gastric cancer risk among *H. pylori* IgG-seropositive individuals [25]. However, some previous studies indicate that excessive alcohol consumption may directly damage the gastric mucosa [26], promote *H. pylori* infection [27], and accelerate the progression of atrophic gastritis.

Among various alcoholic beverages, wine drinkers had the lowest rate of severe gastric atrophy at 21.4%. In contrast, beer drinkers, the largest group by type, showed a high atrophy rate of 44.4% among those consuming 20 g or more of beer per day. These findings are consistent with previous research demonstrating that red wine can prevent *H. pylori*-induced gastric epithelial damage [28]. Additionally, another study indicated that the moderate consumption of wine and beer might help prevent *H. pylori* infection by promoting its eradication [29]. Our results further underscore the nuanced relationship between different types of alcohol and gastric health, suggesting that moderate wine consumption may offer protective effects against gastric atrophy and *H. pylori*-related damage.

Furthermore, Model 3, which included smoking and medication use as covariates, revealed significantly higher atrophy in men. Smoking may influence the progression of atrophic gastritis by impairing blood flow to the gastric mucosa and promoting inflammation [30]. Medications are another factor that may affect the degree of gastric atrophy. Patients in the study were taking a variety of medications, including analgesics, hypertension drugs, anticoagulants, and stomach medications. The use of non-steroidal anti-inflammatory drugs (NSAIDs), warfarin, and aspirin can damage the gastric mucosa, increasing the risk of gastritis and gastric ulceration [31]. Additionally, the long-term use of gastric acid secretion inhibitors may alter the stomach’s microbial environment, contributing to the progression of atrophic gastritis [17].

The limitations of this study are the limited number of patients and the relatively small sample size. Therefore, even the type of medication taken did not withstand analysis and could not be added as a covariate. In addition, the detailed assessment of drinking habits was based on self-reported data and did not consider other lifestyle or genetic factors. Furthermore, patients’ occupations, employment status, and history of gastric problems could not be ascertained. Future studies should examine these factors in more detail.

By clarifying the association between drinking habits and the degree of gastric atrophy in patients treated for *H. pylori* eradication, this study provides valuable insights for patient management and guidance post-eradication treatment. Comprehensive measures, including alcohol consumption control and lifestyle modifications, are particularly important for men and older people.

## 5. Conclusions

Among *H. pylori* eradication patients who were habitual drinkers, gastric atrophy was found to be higher in males and in patients over 60 years of age. In particular, the degree of gastric atrophy tended to be higher in patients who consumed more than 20 g of alcohol per day and who drank more than 5 days per week. Wine drinkers showed the lowest level of atrophy, providing important information for the management of *H. pylori* infection in relation to drinking habits. Future studies are needed to further clarify the impact of drinking habits on gastric mucosal atrophy in *H. pylori*-infected patients through more detailed data collection and analysis and to seek effective intervention methods.

## Figures and Tables

**Table 1 microorganisms-12-01398-t001:** Characteristics of *H. pylori* eradication patients with drinking habits.

Characteristic	n	%
**Sex**		
Male	65	51.2
Female	62	48.8
**Age, years**		
20–29	4	3.2
30–39	25	19.7
40–49	15	11.8
50–59	26	20.5
60–69	34	26.8
70–79	21	16.5
≥80 ha	2	1.6
**Medication status**		
Taking medication	80	63.0
No medication	46	36.2
Unknown	1	0.8
**Type of medication taken (multiple responses)**		
Antihypertensives	16	12.6
Antipsychotic/Antidepressant Medications	12	9.4
Lipid-lowering Agents	11	8.7
Gastrointestinal Medications	11	8.7
Anti-inflammatory and Analgesic Medications	10	7.9
Diabetic Medications	7	5.5
Cardiovascular Medications	5	3.9
Hormonal Agents	3	2.4
Thyroid Medications	3	2.4
Antihistamines	3	2.4
Bone Metabolism Medications	2	1.6
Other Drugs	10	7.9
**Smoking status**		
Smoking (+)	11	8.7
Smoking (−)	116	91.3
**Alcohol consumption per day**		
<20 g	83	65.4
≥20 g	23	18.1
Unknown	21	16.5
**Frequency of drinking**		
4 days or less a week	79	62.2
5 days or more a week	44	34.6
Unknown	4	3.1
**Degree of gastric atrophy**		
C-1	6	4.7
C-2	26	20.5
C-3	27	21.3
O-1	10	7.9
O-2	10	7.9
O-3	13	10.2

Atrophy level: C-1: atrophy near the gastric outlet (pyloric region); C-2: atrophy in the lower 1/3 of the stomach from C-1; C-3: atrophy in the lower 2/3 of the stomach from C-2; O-1: atrophy extends to the perigastric inlet (fenestra); O-2: atrophy extends into the stomach wall (folds seem to disappear); and O-3: total atrophy and loss of folds.

**Table 2 microorganisms-12-01398-t002:** Association between patients’ demographic characteristics and level of gastric atrophy.

	Low Atrophy		High Atrophy		
	n	%	n	%	*p*-Value
**Sex**					
Male	26	51.0	25	49.0	**0.003**
Female	33	80.5	8	19.5	
**Age**					
<60	38	77.5	11	22.5	**0.004**
≥60	21	48.8	22	51.2	
**Alcohol consumption per day**					
<20 g	40	67.8	19	32.2	0.492
≥20 g	10	58.8	7	41.2	
**Frequency of drinking per week**					
4 days or less	36	67.9	17	32.1	0.624
5 days or more	22	62.9	13	37.1	
**Medication status**					
No medication	25	75.8	8	24.2	0.099
Taking medication	34	58.6	24	41.4	

Analysis method: Chi-square test; bold text, statistically significant at *p* < 0.05.

**Table 3 microorganisms-12-01398-t003:** Level of gastric atrophy by type of alcohol.

	Low Atrophy		High Atrophy	
	n	%	n	%
Sake	8	47.0	9	53.0
Beer	37	65.6	20	34.4
<20 g	32	66.7	16	33.3
≥20 g	5	55.6	4	44.4
Shochu	14	63.6	8	36.4
Wine	11	78.6	3	21.4
Average for all subjects	59	64.1	33	35.9

**Table 4 microorganisms-12-01398-t004:** Association between patients’ demographic characteristics and high gastric atrophy.

	Model 1			Model 2			Model 3		
	OR	95%CI	*p*-Value	OR	95%CI	*p*-Value	OR	95%CI	*p*-Value
**Sex**									
Female	1 (Reference)		**0.018**	1 (Reference)		0.071	1 (Reference)		**0.037**
Male	**3.18**	1.22–8.87		2.74	0.92–8.73		**3.43**	1.08–11.99	
**Age**									
<60	1 (Reference)		**0.025**	1 (Reference)		**0.048**	1 (Reference)		0.231
≥60	**2.88**	1.34–7.51		**2.81**	1.01–8.07		2.02	0.64–6.43	

Analysis method: Logistic regression analysis. Notes: Model 1 included sex and age (<60 years and ≥60 years) as explanatory variables. Model 2 was adjusted for alcohol consumption per day (<20 g and ≥20 g). Model 3 was adjusted for alcohol consumption per day (<20 g and ≥20 g), smoking status, and taking medication. Bold text, statistically significant at *p* < 0.05. Abbreviations: OR, odds ratio; CI, confidence interval.

## Data Availability

The data presented in this study are available on request from the corresponding author. The data are not publicly available due to privacy and ethical restrictions, as they have been anonymized in compliance with ethical standards to protect individual privacy.

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
