# Peer review of "Factors Influencing the Degree of Gastric Atrophy in *Helicobacter pylori* Eradication Patients with Drinking Habits"

_microorganisms, 2024, doi:10.3390/microorganisms12071398_

Round 1
Reviewer 1 Report
Comments and Suggestions for Authors
The manuscript is well-written about a novel topic that seeks to explain the association between drinking and gastric atrophy in elderly patients.
Strengths:
- Patients were selected during H. pylori eradication for age and drinking habits.
- An association was found between the type of alcoholic beverage and gastric atrophy. Benefits are provided by wine as compared to other alcoholic beverages.
- Implementation of three models to include different variables in their analysis to determine significant associations.
- Drinking and the type of alcoholic beverages consumed should be part of future epidemiological risk factors when analyzing gastric illnesses in a population.
Weakness:
- No mention of how the patients were identified for H. pylori eradication. Were they positive by urea breath test, CLO test, etc?
- A better description as to the equivalence that >20g or less than 20g to drinks of vodka, sake, or rum, glasses of beer, etc.
- Is occupation or employment part of the questionnaire? Retired or lonely individuals are likely to drink more than active or working individuals.
- Was a history of stomach illnesses part of the questionnaire?
| 45 | |
| hospital for H. pylori eradication, selected only those patients with a drinking habit | 46 |
| among those known to have H. pylori infection, and aimed to examine factors related to | 47 |
| the degree of gastric atrophy in these patients. |
Reviewer 2 Report
Comments and Suggestions for Authors
Ozeki et al. address an important clinical issue regarding the effects of eradicating the pathogenic bacterium Helicobacter pylori on gastric atrophy (and gastric carcinogenesis) in a cohort from the Eastern Asian population. This study provides valuable insights for patient management post-eradication treatment, emphasizing the impact of alcohol consumption and demographic factors on gastric atrophy. The analysis methods and statistics are appropriate, and the findings are clear and solid. However, more details about the specific clinical H. pylori eradication treatments administered to this cohort should be included in the methods, results, or discussion sections, which in fact could be used for perform co-analysis with the other factors.
